# Daily Duration of Eating for Children and Adolescents: A Systematic Review and Meta-Analysis

**DOI:** 10.3390/nu16070993

**Published:** 2024-03-28

**Authors:** Jill Townley, Kate Northstone, Elanor C. Hinton, Julian Hamilton-Shield, Aidan Searle, Sam Leary

**Affiliations:** 1Bristol Dental School, University of Bristol, Dorothy Hodgkin Building, Whitson Street, Bristol BS1 3NY, UK; 2Bristol Medical School, University of Bristol, Oakfield House, Oakfield Grove, Bristol BS8 2BN, UK; kate.northstone@bristol.ac.uk; 3NIHR Bristol Biomedical Research Centre, Diet and Physical Activity Theme, Faculty of Health Sciences, University of Bristol, Education and Research Centre, Upper Maudlin Street, Bristol BS1 3NU, UK; elanor.hinton@bristol.ac.uk (E.C.H.); j.p.h.shield@bristol.ac.uk (J.H.-S.); a.j.searle@bristol.ac.uk (A.S.); 4Bristol Dental School, University of Bristol, 1 Trinity Walk, Bristol BS2 0PT, UK; s.d.leary@bristol.ac.uk

**Keywords:** eating windows, daily eating duration, children, adolescents, chrono-nutrition

## Abstract

Time-limited eating is a dietary intervention whereby eating is limited to a specific window of time during the day. The usual eating windows of adults, and how these can be manipulated for dietary interventions, is well documented. However, there is a paucity of data on eating windows of young people, the manipulation of which may be a useful intervention for reducing obesity. This paper reviewed the existing literature on the eating windows of children and adolescents, aged 5–18 years, plus clock times of first and last intakes and variations by subgroup. Two databases (Medline and Embase) were searched for eligible papers published between February 2013 and February 2023, with forward searching of the citation network of included studies on Web of Science. Articles were screened, and data extracted, in duplicate by two independent reviewers. Ten studies were included, with both observational and experimental designs. Narrative synthesis showed large variations in eating windows with average values ranging from 9.7 h to 16.4 h. Meta-analysis, of five studies, showed a pooled mean daily eating window of 11.3 h (95% CI 11.0, 11.7). Large variations in eating windows exist across different study populations; however, the pooled data suggest that it may be possible to design time-limited eating interventions in paediatric populations aimed at reducing eating windows. Further high-quality research, investigating eating windows and subsequent associations with health outcomes, is needed.

## 1. Introduction

The prevalence of childhood obesity has risen dramatically. In 1975, 4% of 5–19-year-olds worldwide were obese; however, the World Health Organization (2021) reported this had risen to over 18% [1]. Within the UK, 12.2% of children are overweight at age 4–5 years, whilst 9.2% are obese. By 10–11 years of age, this increases to 13.9% and 22.7% overweight and obese, respectively [2], with similar figures seen at 11–15 years [3]. In its simplest terms, obesity is caused by a discrepancy between calorie intake and calorie expenditure; however, the underlying causes of obesity are multi-factorial including genetic, environmental, lifestyle, and cultural factors. Despite these caveats, caloric intake and macronutrient composition play a major role in the development of childhood obesity [4]. Childhood obesity usually persists into adulthood [5] and is associated with increased risks of non-communicable diseases including type 2 diabetes, coronary heart disease, some cancers, stroke, and metabolic syndrome [6]. Alongside physical ill health, childhood obesity is linked to increased risk of depression and anxiety, lower self-esteem, and poorer academic performance [4]. 

Both diet quality and the quantity of intake are established determinants of obesity in all age groups; [4,7] however, recent research has suggested that timing of intake may also have a significant impact [8]. Circadian rhythm refers to the body’s approximately 24-h cycle and includes many physiological processes such as the sleep/wake cycle, hormone secretion, heart rate, and blood pressure. This circadian system is regulated by endogenous circadian clocks, with the suprachiasmatic nucleus in the hypothalamus acting as a master clock regulating peripheral systems [9]. The circadian system has evolved to adapt to seasonal changes, such as change in number of hours of daylight or availability of foods; consequently, external cues such as feed/fasting times or light/dark cycles can cause circadian misalignment [10]. Therefore, restricting times of food intake, thus aligning intake with the circadian clock, may demonstrate beneficial metabolic effects [11]. 

Time-limited eating (TLE) involves consuming all calories within a consistent daily timespan, usually ≤12 h, but without any further restriction on the quality or quantity of foods consumed. TLE regimens vary between 4–12 h of eating, thus inducing a fasting time of 12–20 h [11]. Animal studies have consistently shown TLE interventions reduce body weight and improve metabolic disease risk factors, with reduced total cholesterol, triglycerides, insulin, and glucose levels [12], with similar results seen in adults [12]. A meta-analysis of 19 studies in adults by Moon et al. [13] showed TLE was associated with reduced body weight and total fat mass, and improved cardiometabolic health markers with reduced fasting plasma glucose, systolic blood pressure, and triglycerides. These beneficial metabolic effects are observed independently of weight loss, suggesting that TLE may have intrinsic metabolic effects [11], hypothesised to be via the effect of TLE on circadian rhythms [10]. Whilst TLE is shown to be beneficial in adults it is not known whether it is effective or safe in children or adolescents. To the best of the authors’ knowledge, there are no studies investigating the effects of TLE in paediatric populations with only a few studies investigating feasibility of reducing eating windows (EW) in these populations [14,15,16].

Adults have been shown to have an average daily EW of >12 h [17,18], with erratic eating patterns and variations between weekday and weekend intake [19,20]. However, less is known about the EW of children and adolescents and what factors may influence eating timings. Children have natural inclinations towards specific circadian timings, with the transition from childhood to adolescence known to associate with a shift to a later chronotype [21]. However, detailed knowledge of how this impacts their EW remains unknown. Studies in adults demonstrate differences in mealtimes and dietary patterns across countries [22,23] as well as across socioeconomic groups [24], however, it is not known whether these differences exist in paediatric populations. There remains a need for further research in this area.

In this systematic review we analysed all available evidence from published research to quantify the average daily EW of children and adolescents, aged 5–18 years. Additionally, variations in EW or calorie intake timings by age, nationality, body weight or other subgroups were identified where data allowed. This knowledge can then be used to consider whether it is possible to design interventions based on EW manipulation in children and/or adolescents to improve metabolic health.

## 2. Materials and Methods

This review has been reported according to the Preferred Reporting Items for Systematic reviews and Meta-Analyses (PRISMA-2020) guidelines [25]. The review protocol has been registered on the International Prospective Register of Systematic Reviews (PROSPERO) database (Registration number CRD42023403352). 

### 2.1. Eligibility Criteria 

The review question did not align with the traditional PICO mnemonic for defining eligibility criteria, consequently the CoCoPop (condition, context, population) [26] mnemonic was used. 

#### 2.1.1. Population 

This review included free-living participants aged 5–18 years. Participants with chronic conditions (autism spectrum or other neurodiverse conditions), clinical populations (diabetes, coeliac disease, cystic fibrosis, cerebral palsy, Down’s syndrome, eating disorders, malnourishment, pregnancy, pre- or post-surgery) and elite athletes were excluded. 

#### 2.1.2. Condition 

Participants must have been consuming their habitual dietary intake, assessed via any form of dietary assessment method that included times of intake across a 24-h period. The daily EW, defined as the time from the first to last calorie consumption within each 24-h period, was the primary outcome. Additionally, the clock times of both first and last calorie intakes were reviewed, if data allowed.

#### 2.1.3. Context 

Free-living populations consuming usual intake. Children and adolescents who were hospitalised, or otherwise consuming a modified diet for any reason, were excluded. 

#### 2.1.4. Study Design

Both observational (cohort, case-control, cross-sectional) and experimental (randomised controlled trials, non-randomised trials, crossover trials, feasibility or pilot studies) study designs were included. If experimental studies were included, only the average EWs at baseline, prior to any experimental manipulation, were used. Previous reviews, case studies, and qualitative studies were excluded. Additionally, abstracts where the full-text paper was unavailable were excluded.

### 2.2. Search Strategy 

Searches were designed with support from an information specialist (EP). Two databases (Medline (via OVID) and Embase) were systematically searched in February 2023. Search strategies for each database were similar but adjusted to suit syntax rules for each database. Studies were eligible if published within the last 10 years and in English. Forward citation searching via Web of Science was undertaken for included studies, plus reference lists of included studies were hand-searched. The search strategy used for each database is available in Appendix A.

Studies were uploaded to Endnote reference management software (version 20) for initial de-duplication. Rayyan (https://www.rayyan.ai/, accessed 20 February 2023), was used to screen and manage reports. 

### 2.3. Selection and Extraction

Titles and abstracts were initially screened against pre-specified inclusion criteria by one reviewer (JT). The other members of the review team undertook a blinded second screening (SL, KN, EH, JHS, AS). An over-inclusive approach was taken to this initial screening; if there was any uncertainty about whether abstracts met inclusion criteria they were included for full-text screening. Full texts of relevant reports were retrieved and screened (JT) with blinded duplicate screening (SL, KN, EH, JHS, AS). Any disagreements were resolved through discussion, involving a third reviewer if necessary.

Data extraction forms were developed and piloted using two studies from the literature. Data on study details (study design, sample size, country, participant demographics), method and duration of dietary assessment, and outcomes (duration of daily EW; times of calorie intake) were extracted independently by two reviewers (JT, and SL or KN). Study authors were emailed for missing data when necessary.

### 2.4. Critical Appraisal Assessment

Study quality was assessed using the Joanna Briggs Institute (JBI) critical appraisal checklist for prevalence studies [26], modified for use in this review. This was thought to be the most appropriate tool due to the primary outcome being a single variable (daily EW) rather than a research question assessing associations between exposures and outcomes. Modifications included changing questions 6 and 7 to specify validity and reliability of dietary assessments, and question 8 to specify how EW were derived alongside statistical analysis of these. Guidance documents were modified to reflect these changes. The checklist was then piloted independently using two studies by two reviewers (JT, and SL or KN). Each question was scored (yes = 1, no/unclear = 0), with the total score used to assess overall study quality; no specific a priori cut-off values were used to specify high/low quality [27], instead the results were presented as a table. 

### 2.5. Data Synthesis

Where times of first and last calorie intakes were presented as decimalised times these were converted to clock times for narrative synthesis; e.g., 7.6 h was converted to 7:36 am for ease of interpretation. Daily EWs in hours and minutes were converted to decimalised times to enable statistical analysis. When EW data were missing, mean start and finish times of intake were used to calculate the EW (last intake time-first intake time). Studies using nightly fasting times [28,29] were converted to daily EW using the reciprocal of the median and interquartile range (IQR) given. 

Studies were included in the meta-analysis if they provided a mean EW plus some measure of variation. Stata (version 17) was used for meta-analysis. An inverse-variance random-effects model, using the DerSimonian and Laird method, was used where heterogeneity was high (I^2^ statistic > 50%) [30]. The likelihood of publication bias was not assessed as inspecting funnel plot asymmetry is not recommended when there are less than 10 studies [31].

### 2.6. Sensitivity Analysis

To examine the robustness of the random-effects meta-analysis, an inverse-variance fixed-effects model meta-analysis was also conducted. One study accounted for 97% of the weighting in this fixed-effects sensitivity analysis, consequently, both a fixed and a random effects model were conducted excluding this study.

## 3. Results

### 3.1. Study Characteristics

The literature search identified 6347 papers (Figure 1); 1778 duplicates were removed and 4569 papers underwent title and abstract screening. Of these, 214 papers were identified for full text screening; however, of these, 20 conference abstracts were unavailable as full text reports. Therefore, 194 papers were retrieved and underwent full text screening. In total 10 [15,16,28,29,32,33,34,35,36,37] studies were included in this review, including one study identified from forward searching the citation network of included papers on Web of Science.

Characteristics of included studies are presented in Table 1. The 10 included studies had a total population of 4589 participants, with individual study sample sizes ranging from 22 to 2195. Ages of participants ranged from 6–20 years, with six studies involving adolescents (≥10 years) [16,29,32,33,34,36], and four involving both children (≤9 years) and adolescents [15,28,35,37]. All studies included both sexes, 57.3% of participants were female. Only five studies reported race or ethnicity; 37% of participants were white, 21% black, 21% Hispanic, and 21% other ethnicities. Socioeconomic status was reported by seven studies, measured either by educational level or household income. Study populations were from the USA (n = 4) [15,16,33,37], Brazil (n = 1) [34], India (n = 1) [36], Germany (n = 2) [28,29], The Netherlands (n = 1) [32], and pan-European (n = 1) [35]. Seven studies were observational, with three being cross-sectional [15,34,35], three cohort studies [28,29,36] (with one described as a longitudinal, feasibility study [36]), and one case-control study [32]. Three studies were experimental, with two randomised clinical trials [16,37] (of which one was a pilot study [16]) and one randomised crossover trial [33], all with baseline data available.

### 3.2. Dietary Assessment Methods

Five studies used 24-h recalls to assess dietary intake (Table 2), with four of these using multiple day recalls [16,33,35,37], whilst one was a single recall [34]. Two studies used 3-day weighed food records [28,29], two used single questionnaires [15,32] both of which were designed to assess timing of eating. One study used time-stamped photos via camera phone [36]. Daily EW was defined as the time between first and last meal [35] or eating occasion [33] within any 24-h period, without mention of whether this included all calorie intake events or just food intake. Two studies did not give a definition of how EWs were derived [15,16]. Only four studies specified that the daily eating period [34,36] or nightly fasting duration [28,29] was calculated using all calorie intake events.

### 3.3. First and Last Calorie Intakes

Only four of the included studies [15,32,33,37] gave clock times for first and last calorie intake; of these only one specified that all calorific foods and beverages were used to define intake times [37], the other three specified only food intake (Table 2). Mean first intake times ranged from 07:36 [15] to 09:12 [32], whilst last intake times ranged from 19:39 [33] to 21:32 [32]. One study [32] gave times for weekday and weekend intakes separately, with the weekend having later first and last intake times (weekday: 07:45–20:41, weekend: 09:12–21:32). One crossover study [33] compared intake times for adolescents with experimentally manipulated short vs. healthy sleep durations (with order of sleeping duration randomised). Similar first intake times (08:52 vs. 08:51 for healthy vs. short sleep, respectively) were reported, but later last intake times were reported for the short sleep group (19:39 vs. 20:34, healthy vs. short sleep, respectively).

### 3.4. Duration of Daily Eating

Five studies gave a mean daily EW ranging from 9.7–12.5 h [15,16,33,34,35] (Table 2). Three studies gave a median daily EW [28,29,36]. One stratified by age group, with a median daily EW of 14.23 and 16.36 h in younger (age 13–15 years) and older (age 16–18 years) adolescents, respectively [36]. One was stratified by age and sex, with daily EW increasing with age. In boys, this ranged from 10.97 to 11.88 h in 6–10 and 14–18 year olds, respectively, whilst in girls, it was 11.00 to 11.43 h at age 6–10 and 14–18 years, respectively [28]. Two studies did not give a daily EW, only reporting first and last calorie intake times [32,37] from which EWs were derived. 

Five studies were suitable for meta-analysis [15,16,33,34,35]. Cochran’s Q (95.63, *p* < 0.001) and I^2^ (95.8%) suggested considerable heterogeneity, consequently a random-effects meta-analysis was used. The pooled EW was 11.34 h (95% confidence interval (CI) 10.95, 11.73) (Figure 2).

**Table 2 nutrients-16-00993-t002:** Results of included studies.

Study	Dietary Assessment Method	Mean (SD) First and Last Eating Time (hh:mm)	Daily Eating Window (Hours) (Mean (95%CI))
Berendsen et al. (2020) [32]	Self-reported questionnaire (23 items to assess chrono-nutrition)	Weekdays first: 07:45 (0:51)	Weekdays: 12.93 ^a^
		Weekdays last: 20:41 (1:15)	
		Weekends first: 09:12 (0:47)	Weekends: 12.33 ^a^
		Weekends last: 21:32 (1:11)	
Duraccio et al. (2023) [33]	3 × 24-h recall (weekdays only)	First: 08:52 (1:44)	10.77 (10.34, 11.20) ^d,e^
		Last: 19:39 (1:18)	
Garcez et al. (2021) [34]	Single 24-h recall (day of week not specified)	Data not reported	11.27 (11.14, 11.40) ^d,e^
Intemann et al. (2024) [35]	3 × 24-h recall (only achieved by 23% of participants) (days of week not specified)	Data not reported	11.50 (11.47, 11.53) ^d^
Jain Gupta and Khare (2020) [36]	Photos taken via mobile phone camera for 21 days.	Data not reported	High school (13–15 years): 14.23 ^b^ Higher secondary (16–18 years): 16.36 ^b^
Roßbach et al. (2017) [28]	3-day weighed dietary records (days of week not specified)	Data not reported	Boys 6–10 years: 10.97 (10.22, 11.72) ^c^
			Boys 11–13 years: 11.50 (10.43, 12.30) ^c^
			Boys 14–18 years: 11.88 (10.50, 13.0) ^c^
			Girls 6–10 years: 11.0 (10.17, 11.72) ^c^
			Girls 11–13 years: 11.42 (10.33, 12.33) ^c^
			Girls 14–18 years: 11.43 (10.13, 12.60) ^c^
Roßbach et al. (2018) [29]	3-day weighed dietary records (days of week not specified)	Data not reported	Chronotype T1: 11.12 (10.4, 11.95) ^c^ Chronotype T2: 10.88 (10.10, 12.25) ^c^ Chronotype T3: 10.85 (9.75, 12.37) ^c^
Spaeth et al. (2019) [37]	3 × 24-h recall (2 weekdays and 1 weekend)	First: 08:26 (1:07)	11.45 a
		Last: 19:56 (1:05)	
Tucker et al. (2022) [15]	Self-reported questionnaire (to assess meal and snack patterns, sleep habits, and acceptability of time-limited eating).	Whole sample First: 07:36 (1:30) Last: 20:06 (1:18) Ages 8–10: Not reported Ages 11–17: Not reported	Whole sample: 12.5 (12.24, 12.76) ^d^ Ages 8–10: 12.1 (11.69, 12.51) Ages 11–17: 12.6 (12.30, 12.90) ^e^
Vidmar et al. (2021) [16]	2 × 24-h recall (1 weekday and one weekend)	Data not reported	9.7 (8.79, 10.61) ^d,e^

^a^ Derived (mean last eating time—mean first eating time), ^b^ Median, ^c^ Median (interquartile range), ^d^ Included in meta-analysis, ^e^ Included in subgroup analysis. CI: confidence interval.

### 3.5. Subgroup Analysis—Age

Most studies (6/10) were on adolescents (≥10 years) [16,29,32,33,34,36], whilst four included children (≤9 years) and adolescents [15,28,35,37]. No studies were exclusively on children. Of studies including children and adolescents only two gave daily eating, or nightly fasting durations separately for these age groups [15,28], with neither reporting first and last calorie intake times. Daily EW was reported as 12.10 (11.69, 12.51) and 12.6 (12.30, 12.90) (mean (95% CI)) hours for 8–10-year-olds and 11–17-year-olds, respectively [15]. Median daily EW, stratified by age group and gender, increased with age in both boys and girls [28] (Table 2).

Three of the studies included in the primary meta-analysis were in adolescent-only populations [16,33,34] and one included children and adolescents with results stratified by age group [15]. Subgroup meta-analysis of these adolescent populations increased heterogeneity (Cochran’s Q 88.60, *p* < 0.001, I^2^ 96.61%). The pooled EW in 11–19-year-olds was 11.16 (95% CI 10.27, 12.05) using a random-effects model (Figure 3).

### 3.6. Sensitivity Analysis

The primary meta-analysis was repeated with a fixed-effects model. Findings showed a longer EW of 11.50 h (95% CI 11.48, 11.52) (Appendix A). One study contributed > 97% of the weight for this analysis [35], so the meta-analysis was repeated excluding this study. Heterogeneity remained considerable (Cochran’s Q 95.23 (*p* < 0.001 and I^2^ 96.8%). The mean daily EW was 11.46 h (95% CI 11.36, 11.58) with a fixed-effects meta-analysis and 11.14 h (95% CI 10.28, 12.01) with a random-effects model (Appendix A).

The adolescent subgroup meta-analysis was repeated with a fixed-effects model, with a pooled EW of 11.42 h (95% CI 11.30, 11.54) (Appendix A).

### 3.7. Quality Assessment

Two studies had a critical appraisal score of 6/10, one scored 5/10, two scored 4/10, and five scored 3/10 (Table 3). No studies reported that they used random sampling, whilst two used representative samples and provided details of how sample sizes were chosen [34,35]. However, nine studies provided sufficient detail about their study sampling to assess whether the study sample was a population of interest. Only three studies used valid methods to identify the EW including both weekdays and weekends [16,36,37], whilst nine studies used reliable measurements of the EW [16,28,29,32,33,34,35,36,37]. Four studies specified that the daily EW included all calorie intake events within 24 h [28,29,34,36].

## 4. Discussion

To the best of our knowledge, this is the first review examining the usual daily EW in children and adolescents. The results suggest that children and adolescents have an average EW of 11.3 h each day; however, individual studies showed considerable variation in EW. In adolescent-only populations, this EW was slightly shorter on average at 11.2 h. The results of this subgroup meta-analysis need to be interpreted with caution as only four studies were included in this analysis. Additionally, two of the studies within this subgroup meta-analysis used participants living with obesity from hospital clinics, thus the EW from this may not be generalisable, potentially further limiting this result.

Within the narrative analysis, the three studies that stratified EW by age group showed that EW increased with age, in contrast to the meta-analysis subgroup findings of a shorter EW in adolescents. This may be explained by the large variations in EW seen in this review, alongside the small number of included studies, meaning the limited number of studies may explain the contrasting results. Additionally, the lack of studies giving an EW in children meant it was not possible to examine this statistically, thus it was not possible to see how this varied from an adolescent subgroup. None of the studies investigated why EW may increase as children get older. However, it is possible that this is linked to shorter sleeping durations, thus increasing the time available for eating, with studies showing that children’s sleep durations decline as they get older [38,39,40].

Considerable variation was seen in both the eating times and EW of the included studies. Many factors are known to influence eating in these populations, including cultural norms in relation to meal patterns, with variation seen in both timings of meals in different countries [23] and proportions of foods eaten at different mealtimes [41], which could explain some of the differences seen in these diverse studies. Additionally, other factors such as breakfast skipping [42], night eating [43], snacking [44], and social jetlag (conceptualised as a discrepancy between biological clocks and social commitments, thus affecting sleeping and eating similarly to travel-induced jetlag) [45] are also known to influence intakes, with these factors varying with age groups [46], thus again possibly explaining some of the variation seen.

This review showed large variations in EW between studies, with similar between-study EW variation seen in the adult literature. A large study of 15,000 American adults showed a daily EW of 12.2 h [17], whilst a UK study of 1437 adults showed a median EW of 13.8 h [47]. An Indian study using camera phones to record food and beverage intake events [48] showed extended EW with a median of 15.5 h, similar to the Jain Gupta and Khare study [36] included in this review.

There was evidence of considerable heterogeneity in the meta-analysis, with little overlap in confidence intervals, Cochran’s Q statistic *p* < 0.001, and high I^2^ statistics. Considerable variation was seen in daily EW between individual studies, with several possible methodological and clinical reasons for this high heterogeneity. Firstly, a range of study designs were included within this review; however, there were insufficient studies to be able to perform sub-group meta-analysis based on study design. Secondly, the dietary assessment methods used varied with a combination of retrospective methods, including questionnaires and 24-h recalls, and prospective methods, such as weighed diet records and food photos. Thirdly, studies varied in their definitions of EW with some including all food and beverages, whilst others only specified food intake. Alongside this methodological heterogeneity there was also substantial clinical heterogeneity with variation in ages of populations. Additionally, some studies used populations who were under the care of weight management teams or sleep clinics; these heterogeneous populations may explain differences in EW with participants receiving clinical care potentially differing in habitual dietary intake, either by consuming a reduced-calorie diet or receiving support from trained nutritionists. However, lack of available data prevented further investigation of this issue. This heterogeneity may suggest that combining these studies in a meta-analysis may not be the most appropriate analysis methodology, consequently results should be interpreted with caution.

Whilst the critical appraisal tool used did not allow for categorising studies as low, moderate, or high quality, only two studies had a score of 6/10, with the rest scoring 5/10 or lower, suggesting most were of a lower methodological quality. Only three of the included studies scored a point for using valid dietary assessment methods, defined a priori as multiple day measurements including at least one weekday and one weekend day. The inclusion of weekdays and weekends was considered important due to the volume of existing literature on social jetlag, leading to sleep debt during the school week, often compensated for at weekends [45]. Adolescence brings a marked change toward a later chronotype [49], and therefore social jetlag. The one study that did stratify results by days of the week showed a markedly later first eating time at weekends, consistent with the later wake-up time seen in adolescents at weekends. Additionally, only four of the included studies specified that the calculated EW included all calorie intake events within a 24-h period. The other six studies did not specify how the EW was calculated or only described food intake for the calculation of EW, despite collecting beverage intake data. This suggests a need for consensus as to how an EW is calculated in a rapidly growing field.

The key strength of this review is that it is the first to systematically collate data across a range of ages and across many countries to provide an overview of the daily EW of children and adolescents. However, it has some limitations. Firstly, the broad research question meant that many thousands of studies were identified by the initial searches, consequently limits were applied to reduce these to a manageable level. These limits, particularly the use of an English language filter, may have meant that not all studies were found, however the use of an information specialist to design the searches and the comprehensive process of refining searches meant all efforts were taken for the searches to be exhaustive. Studies were limited to publication within the last 10 years, meaning older studies were not included; however, as children’s eating patterns change over time [50], this review therefore provides current EW, applicable to children and adolescents today. The high heterogeneity seen in the meta-analysis, alongside low numbers of studies, is another limitation of this review, meaning that the results should be interpreted cautiously. Additionally, most studies had low critical appraisal scores further limiting confidence in results. Finally, the planned subgroup analyses based on weight status or nationality were not possible due to lack of available data in the published studies.

The limitations identified here suggest areas for future research. Whilst there is a growing body of evidence on chrono-nutrition in adults, this remains limited in paediatric populations. The limited number of studies identified for this review, alongside the low critical appraisal scores for these included studies, suggests a need for further high-quality research, specifically designed to assess chrono-nutrition in children and adolescents. A lack of valid dietary assessment methods suggests a need for dietary assessment tools to be developed and validated for dietary data collection, including timings of calorie intakes, in children and adolescents. TLE interventions in adults typically limit EW to 8–10 h per day [13], therefore this review suggests there may be scope to design similar interventions in children or adolescents. However, this review suggests that more robust evidence is needed, both to assess current EW in these populations, and to assess the associations of daily EW with health outcomes in children and adolescents, before TLE interventions can be designed for these populations.

## 5. Conclusions

This review demonstrates that both clock times of eating, and daily EW, showed considerable variation between different studies, with EW ranging from 9–16 h. Meta-analysis showed that children and adolescents eat for an average of 11.3 (95% CI 11.0, 11.7) hours each day; however, only five studies were included in this meta-analysis and heterogeneity scores were considerable. Narrative analysis suggested EWs increase as children move into adolescence, however this was not supported by subgroup meta-analysis which showed adolescents had a shorter EW of 11.2 (95% CI 10.3, 12.1) hours. Variation in populations, dietary assessment methods, and definitions of EW may account for some of the differences seen in the narrative analysis and may explain the heterogeneity scores. Critical appraisal scores were relatively low, with no studies scoring more than 6/10. These factors, plus the low number of studies overall, suggest that confidence in the results of this review are low and that further high-quality studies are needed.

## Figures and Tables

**Figure 1 nutrients-16-00993-f001:**
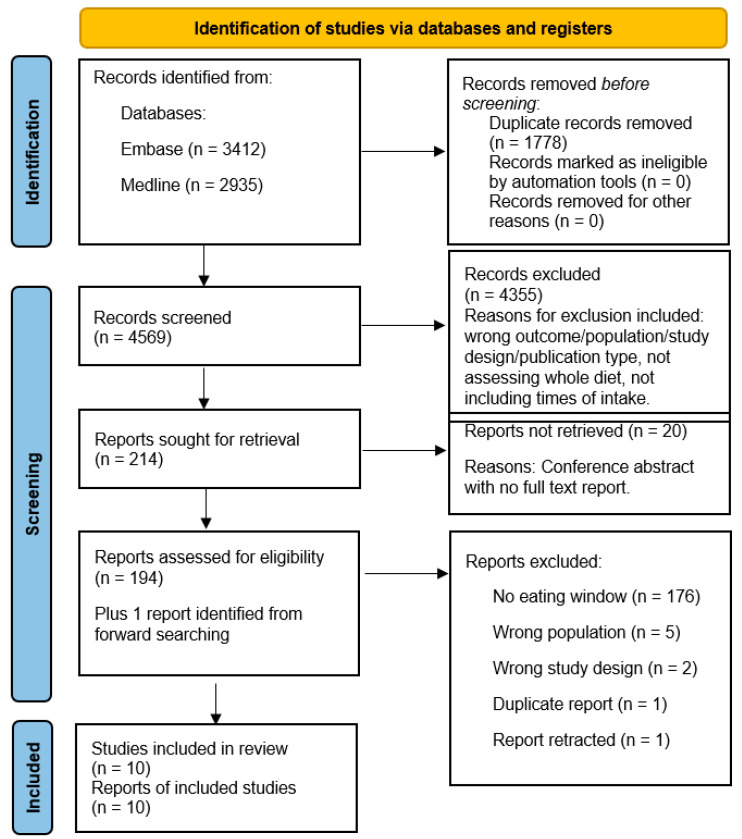
Preferred Reporting Items for Systematic Reviews and Meta-Analyses (PRISMA) flow diagram.

**Figure 2 nutrients-16-00993-f002:**
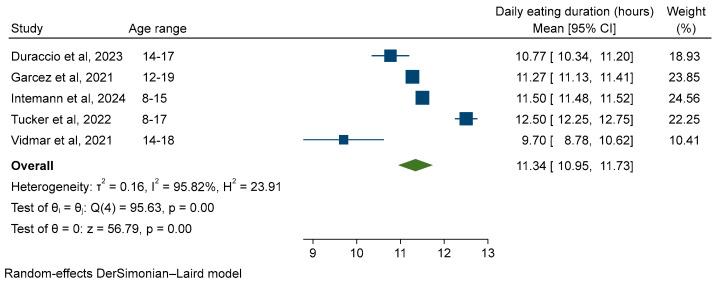
Random-effects meta-analysis of the daily eating window of children and adolescents within a 24-h period. [15,16,33,34,35].

**Figure 3 nutrients-16-00993-f003:**
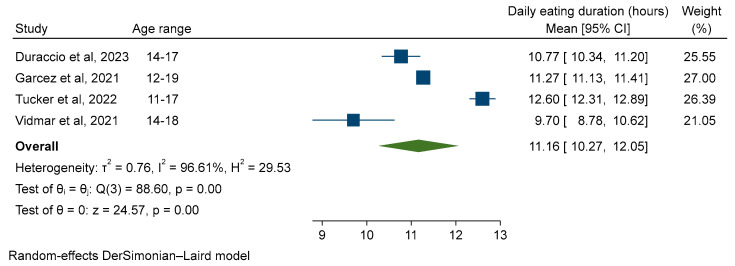
Subgroup analysis. Random-effects meta-analysis of the daily eating window of adolescents aged 11–19 years [15,16,33,34].

**Table 1 nutrients-16-00993-t001:** Characteristics of included studies.

Study	Study Design	Country	Description	Sample Size	Age in Years (Range)	Gender	Race	Ethnicity	Socioeconomic Status	BMI
Berendsen et al. (2020) [32]	Case control	Netherlands	Adolescents with delayed sleep-wake phase disorder and similar aged control group. Data from control group presented.	43	13–20	Female: 56%	Not reported	Not reported	Pre-vocational secondary 14% ^a^ Senior general secondary 49% ^a^ Pre-university 37% ^a^	19.7 (18.7, 22.2) Median (IQR)
Duraccio et al. (2023) [33]	Randomised crossover trial	USA	Short sleep (6.5-h sleep opportunity) or healthy sleep (9.5-h sleep opportunity) for five days with two-day washout period. Healthy sleep data presented	93	14–17	Female: 62%		White 63% Hispanic 2% Black 22% Asian 3% Multiracial 10%	<$50,000 19% ^b^ $50,000–$100,000 32% ^b^ >$100,000 48% ^b^	0.33 (0.91) BMIz Mean (SD)
Garcez et al. (2021) [34]	Cross- sectional	Brazil	Data from 2015 Health Survey of Sao Paulo with Focus in Nutrition (2015 ISA Nutrition).	419	12–19	Female: 47%	White 44% Black 11% Brown 35% Other 5%		≤1: 69% ^c^ >1: 31% ^c^	Overweight 72% (67, 77%) % (95% CI) ^f^
Intemann et al. (2024) [35]	Cross- sectional	Pan-European: Belgium, Cyprus, Estonia, Germany, Hungary, Italy, Spain, Sweden	Data from I.Family study (2013/14).	2195	8–15	Female: 50%	Not reported	Not reported	Low 49% ^d^ High 51% ^d^	0.59 (1.11) BMIz Mean (SD)
Jain Gupta and Khare (2020) [36]	Longitudinal	India	Mobile phone cameras used to collect food data for 21 days in two schools.	13	13–15	Female: 64%	Not reported	Not reported	Not reported	21.6 (16.6–36.9) Median (range)
				9	16–18	Female: 67%				26.0 (15.0–32.4) Median (range)
Roßbach et al. (2017) [28]	Cohort	Germany	Data from Dortmund Nutritional and Anthropometric Longitudinally Designed (DONALD) study, collected 1985–2014.	465	6–10	Male: 100%	Not reported	Not reported	High 62% ^e^	16.0 (15.1, 17.5)
				326	11–13	Male: 100%			High 56% ^e^	18.4 (16.7, 20.4)
				265	14–18	Male: 100%			High 55% ^e^	21.0 (19.0, 23.2)
				461	6–10	Female: 100%			High 61% ^e^	16.0 (14.9, 17.6)
				322	11–13	Female: 100%			High 56% ^e^	18.2 (16.6, 20.6)
				257	14–18	Female: 100%			High 55% ^e^	21.1 (19.1, 23.1) Median (IQR)
Roßbach et al. (2018) [29]	Cohort	Germany	Data from DONALD study with chronotype data, collected 2014–2016.	116 (chronotype T1)	10–18	Female: 49%	Not reported	Not reported	High 89% ^e^	Overweight 8%, Adiposity 3%, Underweight 14% ^g^
				116 (chronotype T2)	10–18	Female: 53%			High 83% ^e^	Overweight 18% Adiposity 1% Underweight 13% ^g^
				114 (chronotype T3)	10–18	Female: 39%			High 81% ^e^	Overweight 18% Adiposity 6% Underweight 7% ^g^
Spaeth et al. (2019) [37]	RCT	USA	Baseline data from an RCT of an intervention to enhance sleep.	87	8–11	Female: 67%	Black 46% White 37% Other 16% Not reported 1%	Hispanic 14% Not Hispanic 86%	Not reported	Normal weight 54% Overweight/obesity 45% Not reported 1% ^h^
Tucker et al. (2022) [15]	Cross-sectional	USA	Survey of parents and children within paediatric weight management clinics to assess timing of eating and acceptability/barriers to time-limited eating.	213 parents/ 159 children	8–17	Female: 57%	Black 22% White 52% Hispanic 16% Other 10%		Not reported	Overweight 3% Obesity 24% Severe obesity 73% ^h^
Vidmar et al. (2021) [16]	Pilot RCT	USA	Baseline data from pilot RCT assessing feasibility, safety, and efficacy of time-limited eating in children undergoing treatment for obesity.	50	14–18	Female: 72%	White 10% Black 6% Asian 8% Hispanic 54% American Indian 4% Mixed race 12%	Hispanic 30% Not Hispanic 64%	Public insurance 74% Annual household income < $50,000 70%	2.30 (0.5) BMIz score Mean (SD)

Socioeconomic status measured by: ^a^ Education level, ^b^ Family annual income ($), ^c^ Per capita family income (minimum wage/month), ^d^ Parental highest educational level (ISCED), ^e^ Maternal education (≥12 years schooling), ^f^ according to WHO (2007) cut off values, ^g^ according to IOTF (2000) BMI cut off values, ^h^ according to CDC (2000) growth charts. BMI: body mass index, IQR: Interquartile range, USA: United States of America, BMIz: body mass index z-score, SD: Standard deviation, CI: confidence interval, RCT: randomised controlled trial, ISCED: International Standard Classification of Education, WHO: World Health Organization, IOTF: International Obesity Task Force, CDC: Centers for Disease Control and Prevention.

**Table 3 nutrients-16-00993-t003:** Quality assessment.

Author, Year	1. Sample Frame	2. Participant Sampling	3. Sample Size	4. Participant and Setting Description	5. Coverage	6. Validity	7. Reliability	8a. Eating Window	8b. Summary Statistics	9. Response Rate	Total
Berendsen et al., 2020 [32]	Yes	No	Unclear	Yes	Unclear	No	Yes	Unclear	Yes	Unclear	4/10
Duraccio et al., 2023 [33]	Yes	No	Unclear	Yes	Unclear	No	Yes	Unclear	Unclear	No	3/10 ^a,b^
Garcez et al., 2021 [34]	Yes	Yes	Yes	Yes	Unclear	No	Yes	Yes	Unclear	Unclear	6/10 ^a,b^
Intemann et al., 2024 [35]	Yes	Yes	Yes	Yes	Unclear	Unclear	Yes	Unclear	Unclear	Unclear	5/10 ^a^
Jain Gupta and Khare, 2020 [36]	Ye	No	Unclear	No	Unclear	Yes	Yes	Yes	Unclear	No	4/10
Roßbach et al., 2017 [28]	No	No	Unclear	Yes	Unclear	No	Yes	Yes	No	Unclear	3/10
Roßbach et al., 2018 [29]	No	No	Unclear	Yes	Unclear	No	Yes	Yes	No	Unclear	3/10
Spaeth et al., 2019 [37]	Unclear	Unclear	Unclear	Yes	Unclear	Yes	Yes	No	No	No	3/10
Tucker et al., 2022 [15]	Yes	Unclear	Unclear	Yes	Unclear	No	No	No	Unclear	Yes	3/10 ^a,b^
Vidmar et al., 2021 [16]	Yes	Yes	No	Yes	Unclear	Yes	Yes	Unclear	Yes	No	6/10 ^a,b^

Questions asked were: 1. Was the sample frame appropriate to address the target population? 2. Were study participants sampled in an appropriate way? 3. Was the sample size adequate? 4. Were the study subjects and the setting described in detail? 5. Was the data analysis conducted with sufficient coverage of the identified sample? 6. Were valid methods used for the identification of the eating window? 7. Was the eating window measured in a standard, reliable way for all participants? 8a. Was the eating window derived appropriately, including first and last calorie intakes in any 24-h period? 8b. Were appropriate summary statistics used? 9. Was the response rate adequate, and if not, was the low response rate managed appropriately? Answer options: Yes, No, Unclear, Not applicable. ^a^ Included in meta-analysis, ^b^ included in subgroup analysis.

## Data Availability

The original contributions presented in the study are included in the article/Appendix A, further inquiries can be directed to the corresponding author.

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
