# Peer review of "Daily Duration of Eating for Children and Adolescents: A Systematic Review and Meta-Analysis"

_nutrients, 2024, doi:10.3390/nu16070993_

Round 1

Reviewer 1 Report

Comments and Suggestions for Authors

In the manuscript, “What is the daily duration of eating for children and adolescents: A systematic review and meta-analysis,” Townley and co-authors address a question that may be an important physiologic factor that impacts obesity among children: “What is the daily duration of eating for children and adolescents?”  This question is important because of the number of interventions being studied and published that propose time limited eating. This paper attempts to determine a baseline value among children by performing a meta-analysis.  The authors take an appropriate methodologic approach to the meta-analysis, with appropriate statistical handling of the heterogeneity including the subgroup analyses and fixed effects model. There is a reasonable discussion of the findings including acknowledgement of limitations, which will be helpful to investigators in planning future research in children/adolescents on eating windows/chrono-nutrition. The meta-analysis ultimately included a small number of studies, but the authors were transparent about eligibility, and a small number is acceptable for the manuscript’s aim (making an observation, not looking for a difference). Moreover, heterogeneity was observed, and coauthors expressed appropriate caution about the low critical appraisal scores of the studies, caution about interpretation of results, and limited generalizability. 

Specific comments/questions for the authors: 

  • Section 3.5 of Results discussed subgroups of age. However, the opening sentence should clearly define the definition of children vs adolescents. While the information is implied later, it would best be defined upfront.  

  • Additional subgroups could be considered for weight status. Some of the studies include subjects of varying ranges of overweight and obesity, some studies appear to be comprised of all subjects with obesity, and some studies include normal weight or underweight subjects. Did the coauthors consider subgroup analyses based on weight status, and why or why not?  

  • In the Discussion the authors mention that some of the subjects are from hospital populations, which could limit generalizability of this study. To take this one step farther, could the heterogeneity of eating windows be related to hospital population status, which may be influenced by obesity status of the subjects?

  • In Table 2, the manuscript citation of reference 36 is incorrect for Jain Gupta and Khare’s paper from 2020. Reference 36 is incomplete in the bibliography, and Gupta Khare’s paper is listed as #37.  

  • In Table 2, the Spaeth paper citation is listed as reference 37, but in the bibliography it is actually number 38.  

  • Subsequent references in the bibliography should be checked and compared to the citation within the manuscript text.  

  • Reference #35 in the bibliography appears to be a submitted paper, but it is in fact now published. The citation appears cut off in the bibliography and should be reloaded.  

Reviewer 2 Report

Comments and Suggestions for Authors

In this paper, available data from published studies were analyzed to quantify the average daily EW of children and adolescents aged 5 to 18 years. The results of this study can be used to develop interventions to improve metabolic health in children and/or adolescents based on influencing EW.

The summary is well formulated and contains all the required elements.

Introduction: the key words used for the literature review were well chosen, as were the inclusion and exclusion factors.

This paper shows that both meal times and daily EW vary considerably between studies, with EW ranging from 9– to 16 hours.

Author Response

Thank you for taking the time to read this manuscript and for your positive comments on this. 

There are no revision points to be addressed. 

Reviewer 3 Report

Comments and Suggestions for Authors

The key strength of this review is that it is the first to systematically collate data across a range of ages and across many countries to provide an overview of the daily EW of children and adolescents. The review was done with great care, the authors described in detail the selection of papers for the study presenting exclusion criteria. Of particular note is Table 3. Quality assessment, which contains much valuable information on the quality and type of studies presented.

Author Response

Thank you for taking the time to read this manuscript and for your positive comments on this.